# Combined Bioinformatic and Experimental Approaches to Analyze miR-182-3p and miR-24-3p Expression and Their Target Genes in Gestational Diabetes Mellitus and Iron Deficiency Anemia During Pregnancy

**DOI:** 10.3390/cimb47080610

**Published:** 2025-08-02

**Authors:** Badr Alzahrani, Bisma Rauff, Aqsa Ikram, Mariya Azam

**Affiliations:** 1Department of Clinical Laboratory Sciences, College of Applied Medical Sciences, Jouf University, Sakaka 72388, Saudi Arabia; baalzahrani@ju.edu.sa; 2Department of Biomedical Engineering, UET Lahore, Narowal Campus, Narowal 51600, Pakistan; bisma.rauff@uet.edu.pk; 3Institute of Molecular Biology and Biotechnology (IMBB), University of Lahore (UOL), Lahore 54000, Pakistan; mariya786azam@gmail.com

**Keywords:** gestational diabetes mellitus (GDM), iron deficiency anemia (IDA), miRNA, pregnancy, bioinformatics, differentially expressed genes (DEGs)

## Abstract

Gestational diabetes mellitus (GDM) and iron deficiency anemia (IDA) are the most common pregnancy-related conditions resulting in adverse maternal and fetal complications. MicroRNAs (miRNAs), particularly miR-182-3p and miR-24-3p, are promising biomarkers as they act as regulatory elements in various diseases; however, their roles in GDM and IDA are unclear. The present study aimed to analyze the expression and functional relevance of miR-182-3p and miR-24-3p in GDM and IDA. Experimental validation via RT-PCR revealed significant upregulation of both miRNAs in GDM and IDA samples. We identified common target genes and signaling pathways associated with these miRNAs, using a combination of data mining, bioinformatic tools (miRDB, TargetScan, miRTarBase, and miRWalk), and differentially expressed gene (DEGs) analysis using the GEO, OMIM, MalaCards, and GeneCards datasets. GO and KEGG pathway analyses revealed that the shared miRNA–mRNA in target genes were enriched in insulin signaling, apoptosis, and inflammatory pathways—key mechanisms implicated in GDM and IDA. Furthermore, hub genes such as IRS1, PIK3CA, CASP3, MAPK7, and PDGFRB were identified, supporting their central role in metabolic dysregulation during pregnancy. These findings demonstrate the potential of miR-182-3p and miR-24-3p as diagnostic biomarkers and therapeutic targets in managing GDM and IDA, offering new insights into the molecular interplay underlying pregnancy complications.

## 1. Introduction

Pregnancy is a dynamic physiological phenomenon that demands precise molecular regulation to support maternal health and fetal development. However, several complications, including gestational diabetes mellitus (GDM) and iron deficiency anemia (IDA), are prevalent and harm the health of mother and child. GDM can lead to an increased risk of neonatal death in affected mothers, compared to non-diabetic mothers [1].

GDM can be defined as glucose intolerance that is first recognized during pregnancy, and can lead to adverse impacts, including macrosomia, preeclampsia, and a high risk of type 2 diabetes later in life [2]. On the other hand, IDA is the most common nutritional disorder during pregnancy and is linked to preterm delivery, low birth weight, and impaired cognitive development in offspring [3]. Evidence suggests a complex association between GDM and IDA during pregnancy. A few studies have reported that IDA is associated with a reduced risk of GDM [4]. However, other studies claim that moderate anemia (especially in the first trimester) may increase the risk of developing GDM later in pregnancy [5,6]. Thus, more research is needed to correlate GDM and IDA in pregnant women.

MicroRNAs (miRNAs) are small non-coding RNA fragments that play a critical role in gene regulation [7]. They appear to be a major contributor to GDM and IDA in pregnant women [7]. Multiple studies have demonstrated that changes in the expression of maternal or placental miRNAs can indicate not only gestational complications, such as preeclampsia, spontaneous abortion, preterm birth, low birth weight, and macrosomia, but also prenatal exposure to environmental pollutants [8,9]. Among these, miR-182-3p and miR-24-3p have emerged as potential regulators of genes involved in metabolic and hematological pathways. Evidence from previous studies indicates that miR-182-3p is linked to tumor initiation and development, largely through its regulatory effect on cell proliferation and migration [10]. Suppression of miR-182-3p mitigates the progression of GDM via regulation of INSR1, indicating its potential as a therapeutic target [11]. It also demonstrates elevated expression in preeclampsia, yet its role in iron deficiency anemia (IDA) has not been clearly defined [12]. Another investigation provided a fresh, integrative perspective on the role of miR-24-3p in the physiological regulation of RBP4, particularly in trophoblast dysfunction and preeclampsia development [13]. Furthermore, elevated levels of miR-24-3p have been detected in women experiencing preeclampsia in the third trimester, but its involvement in IDA among pregnant women is still uncertain. While miR-182-3p and miR-24-3p involvement in GDM has been partially established, their specific functions in IDA during pregnancy remain unclear. Further investigation is needed to determine their diagnostic and therapeutic potential in both conditions.

The present study combines bioinformatic analysis and experimental validation to investigate the expression patterns and regulatory targets of miR-182-3p and miR-24-3p in pregnant women diagnosed with GDM and IDA. By integrating in silico predictions with clinical sample analysis, this study aims to elucidate the molecular mechanisms mediated by miR-182-3p and miR-24-3p, potentially uncovering novel diagnostic biomarkers and therapeutic targets for managing GDM and IDA during pregnancy. We found that miR-182-3p and miR-24-3p play a key role in both GDM and IDA in pregnant women, as they are upregulated in both cases and show a significant correlation with each other. Our findings confirm previously reported results related to diabetes mellitus (DM), thereby validating existing research [14,15]. In contrast, for iron deficiency anemia (IDA), our study generates novel insights, contributing new data to an underexplored area. We also screened out all the genes interacting with the selected miRNA, using online tools. We also analyzed the microarray data of GDM and IDA taken from the GEO database and screened out the differentially expressed genes (DEGs) in GDM and IDA. The overlapping DEGs in the two datasets and those that overlapped with miRNA-associated genes were selected, and functional and pathway enrichment analyses were performed. Finally, the DEGs were uploaded to STRING to construct the protein–protein interaction (PPI) network and identify the hub genes related to GDM. Gene Ontology (GO) analysis revealed that the key genes of GDM, IDA, and selected miRNA are involved in the positive regulation of cysteine-type endopeptidase activity, inflammatory response, organ and bone maturation, post-embryonic development, insulin receptors, and signaling pathways, which are the key factors associated with GDM and IDA in pregnant women. KEGG analysis revealed that insulin resistance, Rap1 signaling, and apoptosis-related pathways are commonly associated with GDM and IDA in pregnant women. Finally, the genes IRS1, MAPK7, PIK3CA, CASP3, PDGFRB, and CLTC were significantly enriched. In this study, the identification of differentially expressed miRNAs and their target genes may contribute to the development of novel biomarkers for the early diagnosis, prognosis, and therapeutic monitoring of GDM and IDA.

## 2. Materials and Methods

### 2.1. Data Mining

Data mining offers a powerful approach to uncover hidden patterns and associations within large biological datasets, providing new insights into disease mechanisms. Here, data mining was performed to identify the role of miR-182-3p and miR-24-3p in GDM and IDA in pregnant women. Understanding the regulatory impact of these miRNAs may aid in identifying novel diagnostic markers or therapeutic targets for managing these conditions.

### 2.2. Expression Profiling of Selected miRNA

Following data analysis, expression analysis of the selected miRNA was performed in a healthy control group and pregnant women with GDM and IDA.

#### 2.2.1. Sample Collection

Thirty gestational diabetes mellitus (GDM) patients, 30 pregnant women with IDA, and 17 healthy pregnant women were included in this research. Each group was analyzed separately as an individual group. Participants were aged between 21 and 35 years to ensure homogeneity and relevance to the study objectives. All participants diagnosed with GDM were in the third trimester (24–28 weeks) of pregnancy. Written informed consent was acquired from all study participants. Patients with a history of pregestational type 1 or type 2 diabetes, chronic kidney disease, thyroid disorders, liver disease, autoimmune disorders, or cancer were excluded. Blood samples were collected from private clinics across various regions of Pakistan, including Lahore, Islamabad, Karachi, and Gujranwala, between January 2024 and March 2025. Peripheral blood (5 mL) was drawn using EDTA vacutainers. Serum was immediately separated and stored in plastic vials at −80 °C. This study was approved by the Institutional Ethics Committee of the Institute of Molecular Biology and Biotechnology (IMBB) of The University of Lahore (Project: PR/AI/GDM-1, Date 10 January 2025).

#### 2.2.2. RNA Extraction

A Quick-RNA Kit (Zymo) was used to extract total RNA from the serum samples, following the manufacturer’s instructions.

#### 2.2.3. RT-PCR

cDNA was synthesized from total RNA, using a cDNA Synthesis Kit (Thermo Scientific #K1622, Waltham, MA, USA). The synthesized cDNA was used for RT-PCR analysis, using an RT-PCR kit (Thermo Scientific #K0221). The gene encoding GAPDH was used as a housekeeping gene. The reaction mixture was prepared by mixing 12.5 µL of Maxima SYBR Green/ROX qPCR Master Mix (2X), 1 µL of forward primer, 1 µL of reverse primer, 2 µL of the template DNA (≤500 ng), and 8.5 µL of nuclease-free H_2_O. The procedure comprised 40–45 cycles, performed under the following conditions: initial denaturation at 95 °C for 5 min, denaturation at 95 °C for 20 s, annealing at 49–62 °C for 30 s for non-coding RNAs, and extension at 72 °C for 45 s. Healthy control samples were also used for each run. The ΔΔCT method was used to determine the relative expression levels of miR-182-3p and miR-24-3p.

### 2.3. Statistical Analysis

GraphPad Prism 8.0.2 was used for the statistical analysis. A one-way analysis of variance (ANOVA) and Tukey’s multiple comparison tests were performed to identify significant differences.

### 2.4. Prediction of miRNA Targets

The targets of the selected miRNAs were predicted using four online databases: miRDB [16], TargetScan [17], miRWalk [18], and miRTarBase [19]. Targets that were concurrently identified by at least three of these databases were selected for further analysis.

### 2.5. Identification of Gene Expression Profile and Hub Genes

The GSE154413 dataset was obtained from the Gene Expression Omnibus. This dataset was collected from placental samples from eight GDM and five control patients. Genes that were differentially expressed across experimental conditions were identified. R (version 3.6.3) was used to analyze gene expression. The DESeq package was employed to detect DEGs with strict thresholds (log2FC ≥ 1 and *p* ≤ 0.05). Thorough data mining was performed to screen out all genes reported against IDA. In addition, the GeneCards [20], MalaCards [21], and OMIM [22] datasets were employed to screen out all genes associated with IDA in more than 12,000 studies. Genes common in both the selected miRNA-interacting genes and the datasets were included in the subsequent analysis. These genes (i.e., those overlapping between miRNAs and DEGs) were selected for PPI network construction and to identify the common hub genes. The STRING database was also used to build the PPI network (https://string-db.org/, accessed on 24 July 2025). Cytoscape (version 3.10.3) was employed to visualize the PPI network. Finally, we selected the top 10 genes as hub genes, using the cytoHubba plug-in and the number of degrees as the standard.

## 3. Results

### 3.1. Data Analysis

Data analysis was performed to identify all possible roles of the selected miRNA in GDM and IDA in pregnant women. miR-182-3p is involved in cardiovascular diseases, cancers, and autoimmune diseases, but its exact role in GDM and IDA in pregnant women is still unknown (Figure 1). Likewise, miR-24-3p is involved in bone, cardiovascular, renal, brain, and viral diseases, cancers, and immune responses; however, its role in GDM and IDA in pregnant women is still unclear (Figure 1).

Figure 1 illustrates the involvement of miR-182-3p and miR-24-3p in human diseases, based on integrated data mining, using publicly available datasets (PubMed and Google Scholar). This figure highlights key disease categories, including cardiovascular diseases; cancers; autoimmune disorders; and bone, brain, renal, and viral diseases, where these miRNAs are differentially expressed or functionally implicated.

### 3.2. Sample Collection

A case-control study was conducted to quantify the expression profiles of miR-182-3p and miR-24-3p in pregnant women diagnosed with gestational diabetes mellitus (GDM) (n = 30), iron deficiency anemia (IDA) (n = 30), and healthy pregnant controls (n = 17), all within the age range of 17 to 35 years. The control group consisted of 17 healthy pregnant women aged 17–35 years (Table 1).

### 3.3. Expression Analysis of Selected miRNA in GDM

As shown in Figure 2A, the GDM group exhibited significantly (* *p* < 0.001) higher miR-182-3p expression than the control. Furthermore, miR-24-3p expression was also higher (** *p* < 0.008) in the GDM group than in the control (Figure 2B).

### 3.4. Expression Analysis of Selected miRNA in Pregnant Women with IDA

As shown in Figure 2C, miR-182-3p demonstrated a significantly (*** *p* < 0.001) higher expression in the GDM group than in the control. Likewise, miR-24-3p expression was significantly (*** *p* < 0.002) higher in GDM patients than in the control (Figure 2D).

### 3.5. Comparison of the Relative Expression of Selected miRNA in Pregnant Women with GDM and IDA

The relative expression patterns of miR-182-3p and miR-24-3p in pregnant women with GDM and IDA were compared using one-way ANOVA with Tukey’s multiple comparison test. A significant relation (*p* < 0.0001) between miR-182-3p and miR-24-3p was observed in pregnant women with GDM and IDA (Figure 2D,E).

### 3.6. Exploring miRNA Gene Targets

Several bioinformatics tools are used to predict miRNA–mRNA interactions based on sequence pairing, evolutionary conservation, and thermodynamic stability. Here, we explored all possible genes interacting with miR-182-3p and miR-24-3p, using online bioinformatic tools. Gene targets identified using miRDB, TargetScan, miRwalk, and miRTarbase were selected. After removing the duplicates in miR-182-3p and miR-24-3p, a total of 1347 (388 + 959) genes were chosen for further analysis.

### 3.7. Analysis of Gene Expression Profile Data

All DEGs were considered and analyzed in the microarray, considering *p* > 0.05 and the logFC value. GEO2R was used to identify and screen out all DEGs between GDM patients and the control (Figure 3). Data refinement procedures were performed to improve the accuracy of the findings: (1) DEG sets without the corresponding gene symbols were excluded; (2) DEGs with logFC values > 1 and *p* < 0.05 were included. These refinement procedures ensured the identification of DEGs with meaningful biological activities. Volcano plots were constructed based on these analyses (Figure 3). We did not find any studies related to IDA in pregnant women in the GSE dataset. Therefore, we included all genes associated with IDA through data mining to enhance the relevance of our study to human targets. In particular, we employed GeneCards, MalaCards, and the OMIM dataset, which collectively report IDA-specific genes based on more than 12,000 published studies.

### 3.8. PPI Network and Hub Genes

All DEG common in all datasets and miRNA-interacting genes were selected for further analysis. A total of 351 genes were found to be common in all selected databases and miRNAs. These genes were further selected for STRING PPI network and hub gene identification. GO functional analysis revealed that these targets are involved in the positive regulation of cysteine-type endopeptidase activity, organ and bone maturation, and related processes (Figure 2). KEGG pathway enrichment analysis was conducted to identify relevant signaling pathways between GDM and IDA in pregnant women. Most genes participated in the following pathways: Rap1 signaling (34), insulin resistance (32), apoptosis (45), and Herpes simplex virus 1 infection (32) (Figure 4). Finally, hub gene identification revealed that genes p53, IRS1, PIK3CA, CASP3, PDGFRB, and MAPK7 were significantly enriched (Figure 5).

## 4. Discussion

miRNA plays a key role in gene expression regulation. Various studies have reported that miRNAs could be ideal biomarkers for human diseases, but there are still knowledge gaps in this area. For example, the roles of miR-182-3p and miR-24-3p in GDM and IDA in pregnant women are unclear. Therefore, the present study aimed to identify the association between these diseases and selected miRNAs through data mining. Further, expression analysis of the selected miRNA in GDM and IDA was evaluated using RT-PCR. Finally, integrating database (miRDB, TargetScan, miRTarBase, and miRWalk) predictions with DEG analysis using the GEO dataset enhanced the robustness of our findings.

Our data mining revealed that the selected miRNAs (miR-182-3p and miR-24-3p) played a key role in numerous diseases, including cancer, cardiovascular, autoimmune, renal, and viral diseases. miR-182-3p has been associated with glucose metabolism and insulin resistance [9], which aligns with its altered expression in our GDM cohort. Our analyses are grounded in the observation that inhibition of miR-182-3p is associated with an increased risk of GDM and adverse pregnancy outcomes. On the other hand, miR-24-3p has been implicated with type 2 diabetes in patients with previous GDM [15]. While these miRNAs have been partially characterized in GDM, their specific functions in IDA are still unknown, highlighting the need for further research to explore their diagnostic and therapeutic potential in both conditions. Our experimental validation revealed significant upregulation of selected miRNAs in both GDM and IDA samples, supporting their proposed role in impairing insulin signaling pathways. Our results align with previous findings in diabetes mellitus (DM), reinforcing established research, while offering new insights into iron deficiency anemia (IDA), an area with limited existing data.

The concordance between bioinformatic predictions and experimental data strengthens the proposed role of the selected miRNAs as critical regulators in pregnancy complications. Using a set of bioinformatic tools, we identified target genes for miR-182-3p and miR-24-3p that were previously reported in relation to various metabolic and hematological disorders [23]. We also analyzed the DEGs among GDM patients through GEO analysis. Although no studies directly examining IDA in pregnant women were identified in the GSE dataset, we broadened our approach to enhance the applicability of our findings to human disease. Hence, we conducted comprehensive data mining to compile a list of genes associated with IDA, drawing from GeneCards, MalaCards, and the OMIM database. These resources collectively integrate gene–disease associations based on evidence from more than 12,000 scientific publications. All genes common in both miRNA targets and DEGs of IDA and GDB were selected for further analysis. GO analysis revealed that the key genes are involved in the positive regulation of cysteine-type endopeptidase activity, inflammatory response, organ and bone maturation, post-embryonic development, and insulin receptor and signaling pathways, which are key factors associated with GDM and IDA [24,25]. KEGG pathway analysis was performed to identify the crucial signaling pathways. The most associated pathways included insulin resistance, Rap1 signaling, and apoptosis, which are commonly associated with GDM and IDA in pregnant women. This finding indicates that the common genes of GDM, IDA, and selected miRNA may be closely related and play an important role in the complications associated with GDM and IDA.

Furthermore, we determined that the hub genes of GDM, IDA, and the selected miRNA play an important role in various metabolic pathways. Genes p53, IRS1, MAPK7, PIK3CA, CASP3, and PDGFRB were significantly enriched. p53 is highly expressed in GDM and activates the JAK/STAT signaling pathway [26]. In addition, p53 participates in the regulation of metabolism, particularly glucose metabolism [27]. Recent investigations have revealed the pivotal function of p53 in iron regulation [28]. On the other hand, IRS1 is a key factor in insulin signaling and is involved in GDM [29]. A study reported that IRS1-deficient mice exhibit insulin sensitivity, proving its role in the insulin pathway [30]. In addition, iron metabolism plays a crucial role in the insulin pathway [31]. A high glucose concentration activates the expression of PIK3CA [32]. The PI3K/Akt pathway contributes to insulin metabolism and is strongly associated with the regulation of placental development [33]. Furthermore, the deregulation of caspase-3 in the placenta of GDM patients can lead to abnormal apoptosis [34]. A study reported that the PDGF/PDGFR signaling pathway plays an important role in the onset and progression of diabetes and its related complications [35]. However, this pathway is also involved in normal embryonic development and cellular differentiation. On the other hand, the MAPK7 pathway is involved in diabetes development. This is expected because this pathway plays a role in the insulin signaling pathway by regulating glucose hemostasis. Our findings contribute to the growing understanding of how miRNAs regulate key pathways during pregnancy complications and suggest potential diagnostic and therapeutic avenues. This analysis offers several notable strengths. These findings pave the way for future research on miRNA-based diagnostic tools and therapeutic strategies aimed at improving maternal and fetal outcomes. A key strength of this study lies in its integrative approach, combining both bioinformatic analyses and experimental validation to investigate the roles of miR-182-3p and miR-24-3p in two prevalent pregnancy-related conditions—gestational diabetes mellitus (GDM) and iron deficiency anemia (IDA). This dual methodology enhances the reliability and biological relevance of the findings. Further, Experimental validation of microRNA expression through qRT-PCR in patient samples adds clinical significance and strengthens the translational potential of the data. Despite offering valuable insights, this study has several limitations that should be acknowledged. First, while the integration of bioinformatic and experimental approaches strengthens the findings, the study lacks in vivo functional validation, which is essential to fully understand the biological consequences of miR-182-3p and miR-24-3p dysregulation in the context of gestational diabetes mellitus (GDM) and iron deficiency anemia (IDA). Without experimental models or patient-derived functional assays, it remains unclear how these miRNAs mechanistically contribute to the onset or progression of these conditions. While target genes are analyzed, the precise downstream molecular and cellular consequences of miRNA dysregulation in vivo require more in-depth investigation. Ultimately, comprehensive functional experiments are essential to clarify the precise roles of these genes and miRNAs in the development and progression of IDM and GDM. To strengthen the current evidence, future research should employ expanded cohorts, time-course analyses, and in-depth experimental validation

In conclusion, our study demonstrated the role of miR-182-3p and miR-24-3p as regulatory molecules in GDM and IDA during pregnancy. The study targets miR-182-3p and miR-24-3p—miRNAs with known disease relevance—providing specific insights into their roles in GDM and IDA, two major pregnancy complications.

## Figures and Tables

**Figure 1 cimb-47-00610-f001:**
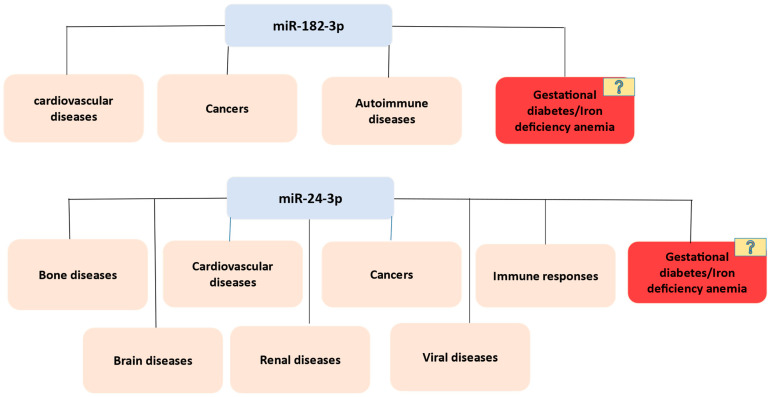
Roles of miR-182-3p and miR-24-3p in human diseases, identified through data mining.

**Figure 2 cimb-47-00610-f002:**
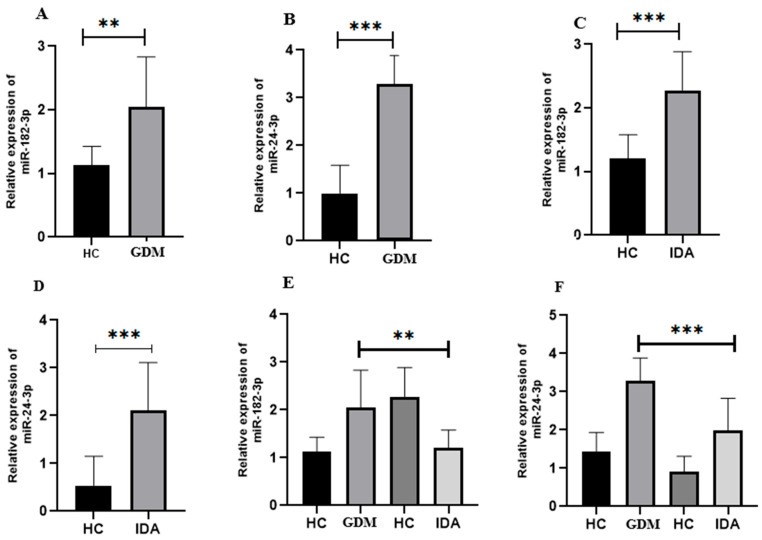
Expression analysis of miR-182-3p and miR-24-3p in GDM and IDA. (**A**) miR-182-3p expression level in the control (healthy) and GDM patients; (**B**) miR-24-3p expression level in the control and GDM patients; (**C**) miR-182-3p expression pattern in the control and IDA patients; (**D**) miR-24-3p expression pattern in the control and IDA patients; (**E**) relative expression of miR-182-3p in GDM and IDA patients; (**F**) comparative expression of miR-24-3p in GDM and IDA patients. Student’s *t*-test was used to compare the average values and variability between two normally distributed data sets. One-way ANOVA was applied for comparisons involving three or more such groups. Significance was set at ** *p* < 0.01, and *** *p* < 0.001.

**Figure 3 cimb-47-00610-f003:**
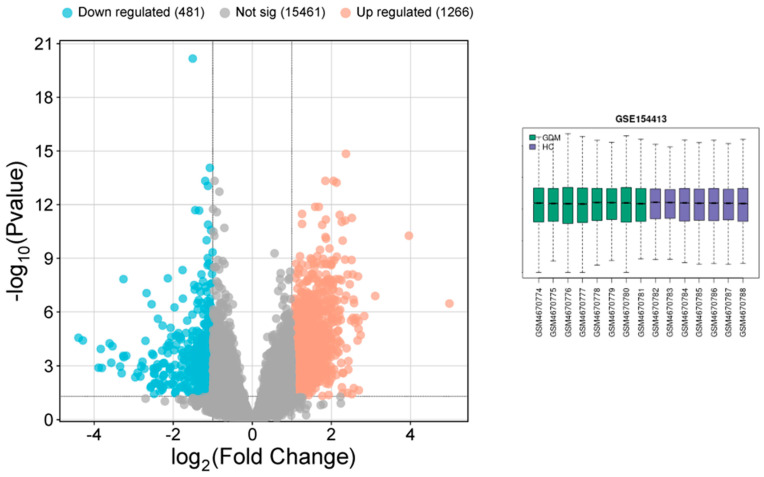
Volcano plots illustrating the differentially expressed genes (DEGs) in GSE154413 (GDM).

**Figure 4 cimb-47-00610-f004:**
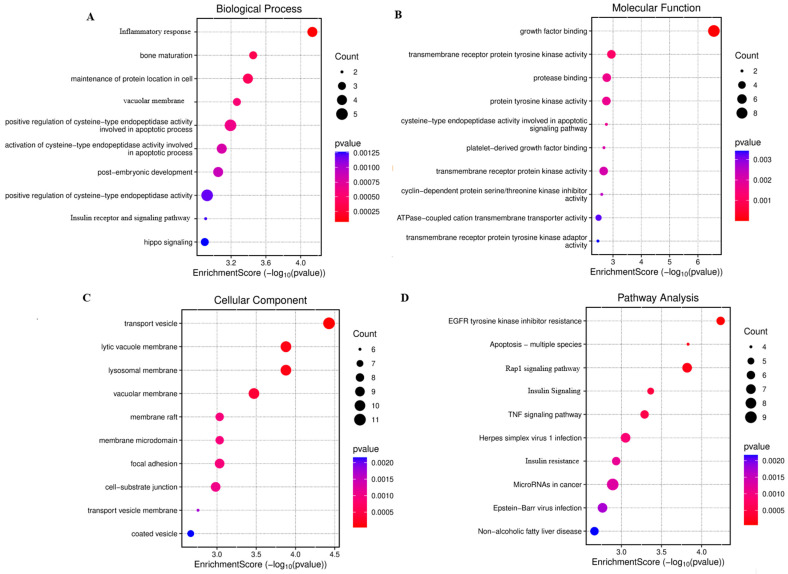
Functional annotation and enriched pathway analysis, visualized using a bubble plot. (**A**) Biological-process-related GO terms; (**B**) molecular-process-related GO terms; (**C**) cellular-process-related GO terms; (**D**) KEGG pathway analysis.

**Figure 5 cimb-47-00610-f005:**
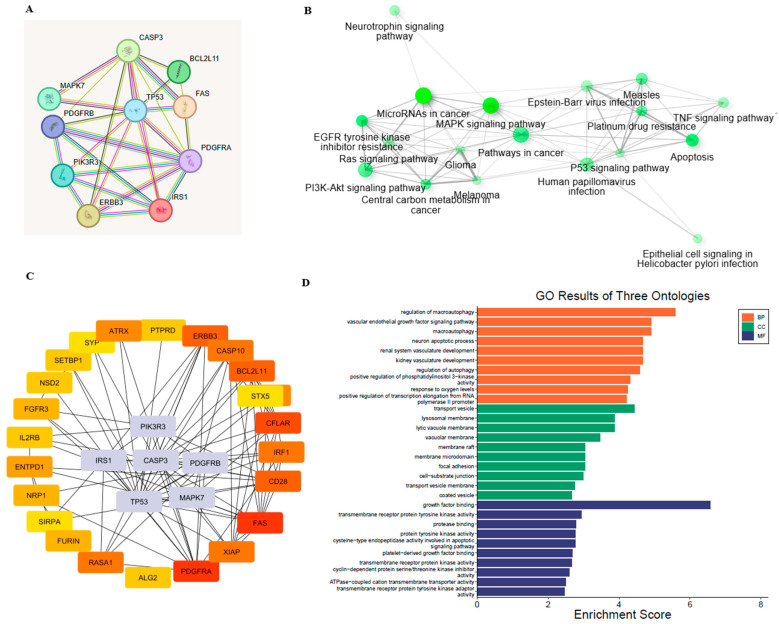
(**A**) Protein–protein interaction (PPI) network of key genes constructed using the STRING database. Nodes represent proteins; edges represent predicted functional associations. Stronger associations are indicated by thicker lines. (**B**) KEGG pathway enrichment network showing key biological pathways associated with the gene set. (**C**) Hub gene subnetwork derived from the PPI network, using cytoHubba. Central hub genes are shown with increased node size and intensity, whereas first-degree neighbors are color-coded by interaction strength. (**D**) GO enrichment results categorized into biological process (BP), cellular component (CC), and molecular function (MF) ontologies.

**Table 1 cimb-47-00610-t001:** Demographic details of the study population.

Variables	G1 (Control)	G2 (GDM)	G3 (IDA)
**Age**	**N = 17**	**N = 30**	**N = 30**
Mean	25.35	29.5	24.57
SD	4.513	3.014	3.234
SEM	1.095	0.5503	0.509
Range	16	11	11
**Gestational week**			
Mean	25.67	26.13	26.6
SD	1.455	1.432	1.276
SEM	0.343	0.2614	0.2329
Range	4	4	4

## Data Availability

The data that support the findings of this study are available from the corresponding author upon reasonable request.

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
