# Peer review of "Combined Bioinformatic and Experimental Approaches to Analyze miR-182-3p and miR-24-3p Expression and Their Target Genes in Gestational Diabetes Mellitus and Iron Deficiency Anemia During Pregnancy"

_cimb, 2025, doi:10.3390/cimb47080610_

Round 1

Reviewer 1 Report

Comments and Suggestions for Authors

In this manuscript the authors examined the role of miR-182-3p and miR-24-3p in GDM and IDA.  There are serious problems with the study and manuscript.

  1. The authors looked at the expression of miR-182-3p and miR-24-3p in the blood samples of patients and controls by qPCR.  What cell types are not determined, maybe some immune cells.  Then they looked at GSE datasets, which were from different cell types/tissues.  The relationship between the two is unclear, so any conclusion is suspect.
  2. The manuscript mentioned GSE154414 and GSE46287 in 2.6, GSE9984 and GSE103552 in 3, then GSE154413 and GSE46287 in Figure 3. Which datasets did they analyze?
  3. The writing is mixed up. For example, 2.1 is not a Materials and Methods content.  RT-qPCR procedures were not adequately explained.  Figure 1 is in an odd place and removable.
  4. Figure 2 results are hard to assess without the raw data and control miRNAs whose expression is constant. The miR-182-3p and miR-24-3p differences between the controls and patients are much larger than usual.  It is almost an absent vs present contrast, or comparison between different cell types.

Author Response

Response letter:

Dear Editor,

We wish to express our appreciation for your in-depth comments, suggestions, and corrections, and we would like to convey our sincere thanks for allowing us to improve our manuscript entitled “Combined Bioinformatic and Experimental Approaches to Analyze miR-182-3p and miR-24-3p Expression and Their Target Genes in Gestational Diabetes Mellitus and Iron Deficiency Anemia during pregnancy”.

We greatly appreciate the reviewers’ efforts in carefully reviewing our paper and for offering salient suggestions, which were of great help to us during this revision. We worked hard to be responsive to them.

We believe that we exhaustively resolved all points raised by the referees. We hope that the esteemed referees will find our responses satisfactory, and we are willing to include any future suggestions that the referees might have. Below is an abridged summary of each of the comments with a detailed response. We hope our revision meets your approval. Should you find the paper requires further clarification or revision, we most certainly stand ready to do so.

Looking forward to your positive response

Sincerely,

Aqsa Ikram (PhD)

Assistant Professor

Institute of Molecular Biology and Biotechnology,

University of Lahore.

Comment:

The authors looked at the expression of miR-182-3p and miR-24-3p in the blood samples of patients and controls by qPCR. What cell types are not determined, maybe some immune cells.

Response: we are thankful to the reviewer for his valuable comment. We have now corrected our mistake in the revised manuscript. For the miRNA expression analysis of selected miRNA we had taken serum not whole blood.

Comment:

Then they looked at GSE datasets, which were from different cell types/tissues. The relationship between the two is unclear, so any conclusion is suspect.

Response: We are thankful to the reviewer for his insightful comment. In the GSE dataset, we did not find any study specifically related to IDA in pregnant women. Therefore, we initially used a rat model of IDA. However, to enhance the relevance of our study to human targets, we have revised the manuscript to include all genes associated with IDA through data mining. Specifically, we utilized multiple tools, including GeneCards, MalaCards, and the OMIM dataset, which collectively report IDA-specific genes based on more than 12,000 published studies.

Comment:

The manuscript mentioned GSE154414 and GSE46287 in 2.6, GSE9984 and GSE103552 in 3, then GSE154413 and GSE46287 in Figure 3. Which datasets did they analyze?

Response:

We are thankful to the reviewer for his valuable comment. In the revised manuscript we have now corrected our mistake. In the revised manuscript we have now use only GSE154414 dataset.

Comment:

The writing is mixed up. For example, 2.1 is not a Materials and Methods content.

Response:

We are thankful for the reviewer comment. We have now corrected our mistake in the revised manuscript.

Comment:

RT-qPCR procedures were not adequately explained.

Response:

We are thankful to the reviewer. We have no added thedetails of RT PCR prcedure in detail in the revised manuscript and is also copied below for your consideration.

RT-PCR

From the total RNA, cDNA was synthesized by using cDNA Synthesis Kit (Thermo Scientific #K1622). The synthesized cDNA was used for RT-PCR by using RT-PCR kit (Thermo Scientific #K0221). As a housekeeping gene, GAPDH was used. The reaction mixture was prepared by adding the 12.5µl of Maxima SYBR Green/ROX qPCR Master Mix (2X), 1µl of forward primer, 1µl of the reverse primer, 2µl of the template DNA (≤500ng) and 8.5µl nuclease free H2O. The procedure of 40-45 cycles was performed at the conditions; initial denaturation for 5min at 95ᵒC, denaturation for 20sec at 95ᵒC, annealing range of all non-coding RNAs falling between 49ᵒC-62ᵒC for 30 sec, and extension temperature for 45sec 72ᵒC. Samples of non-treated groups as a healthy control were also used for each run. The ΔΔCT method was used to find out the relative expression levels of miR-182-3p and miR-24-3p.

Comment:

Figure 1 is in an odd place and removable.

Response:

Thank you for your observation. We have now modify the figure 1.

Comment:

Figure 2 results are hard to assess without the raw data and control miRNAs whose expression is constant. The miR-182-3p and miR-24-3p differences between the controls and patients are much larger than usual. It is almost an absent vs present contrast, or comparison between different cell types.

Response:

We appreciate the reviewer’s insightful comment. In response, we have consulted RNA expression analysis and biostatistics experts and re-evaluated our findings by conducting experiments in triplicates. Figure 2 has been accordingly revised.

Reviewer 2 Report

Comments and Suggestions for Authors

First of all, these miRNAs have already been linked to Gestational Diabetes Mellitus, therefore these results are not novel. This manuscript is poor as there is only one figure that represents actual bench work, and the rest are bioinformatic results that basically are automatically generated on platforms. The results cannot be translated to woman as the bioinformatic work was performed on a rat database.

Please expand and pave the way on your introduction, to explain why you chose this methodology as I do not see the link between iron deficiency during pregnancy and a non-pregnant iron deficient rat database, with four groups of 3 rats each. At the beginning it is stated that this paper was going to be about pregnancy women, but suddenly in the materials and methods RATS come in, until page 3.

A lot of information is missing regarding the human samples collected, for example:  from which clinic were they collected?, from which period?, was informed consent obtained?, at what trimester of pregnancy they were?.  This information is currently misplaced in the results sections, whereas it should be on the materials and methods section.

Also, demographics are not described for the patient’s samples.

The introduction needs to be improved. At the moment it doesn’t flow. Organize the ideas. Also include other diseases during pregnancy for which there is already evidence, for instance ectopic pregnancy and preeclampsia.

Data analysis needs to be improved as several graphs need to be analyzed by t-test, not by ANOVA

** PLEASE REVISE THE ATTACHED FILE***

There are missing crucial references that need to be discussed such as:

https://onlinelibrary.wiley.com/doi/10.1155/2017/8067972

https://pubmed.ncbi.nlm.nih.gov/38404088/

https://www.frontiersin.org/journals/endocrinology/articles/10.3389/fendo.2022.892587/full

Comments on the Quality of English Language

In general, the document is full of grammar mistakes, duplicated words, extra spaces. Formatting is required throughout the document, there are several Font types, sizes and even colors. All through the document there are several sentences with a mix or plural and singular, such as “selected miRNAs played significant role” (significant roles OR a significant role). Remove bold text throughout.

Author Response

Dear Editor,

We wish to express our appreciation for your in-depth comments, suggestions, and corrections, and we would like to convey our sincere thanks for allowing us to improve our manuscript entitled “Combined Bioinformatic and Experimental Approaches to Analyze miR-182-3p and miR-24-3p Expression and Their Target Genes in Gestational Diabetes Mellitus and Iron Deficiency Anemia during pregnancy”.

We greatly appreciate the reviewers’ efforts in carefully reviewing our paper and for offering salient suggestions, which were of great help to us during this revision. We worked hard to be responsive to them.

We believe that we exhaustively resolved all points raised by the referees. We hope that the esteemed referees will find our responses satisfactory, and we are willing to include any future suggestions that the referees might have. Below is an abridged summary of each of the comments with a detailed response. We hope our revision meets your approval. Should you find the paper requires further clarification or revision, we most certainly stand ready to do so.

Looking forward to your positive response

Sincerely,

Aqsa Ikram (PhD)

Assistant Professor

Institute of Molecular Biology and Biotechnology,

University of Lahore.

Comment:

The results cannot be translated to woman as the bioinformatic work was performed on a rat database.

Response:

We are thankful to the reviewer for his insightful comment. In the GSE dataset, we did not find any studies specifically related to IDA in pregnant women. Therefore, we initially used a rat model of IDA. However, to enhance the relevance of our study to human targets, we have revised the manuscript to include all genes associated with IDA through data mining. Specifically, we utilized multiple tools, including GeneCards, MalaCards, and the OMIM dataset, which collectively report IDA-specific genes based on more than 12,000 published studies.

Comment:

Please expand and pave the way on your introduction, to explain why you chose this methodology as I do not see the link between iron deficiency during pregnancy and a non-pregnant iron deficient rat database, with four groups of 3 rats each. At the beginning it is stated that this paper was going to be about pregnancy women, but suddenly in the materials and methods RATS come in, until page 3.

Response:

We appreciate the reviewer’s constructive suggestion. We have now modified the introduction part to show the importance and relevance of the study. We have also modified the methodology part. In the GSE dataset, we did not find any studies specifically related to IDA in pregnant women. Therefore, we initially used a rat model of IDA. However, to enhance the relevance of our study to human targets, we have revised the manuscript to include all genes associated with IDA through data mining. Specifically, we utilized multiple tools, including GeneCards, MalaCards, and the OMIM dataset, which collectively report IDA-specific genes based on more than 12,000 published studies.

Comment:

A lot of information is missing regarding the human samples collected, for example: from which clinic were they collected?, from which period?, was informed consent obtained?, at what trimester of pregnancy they were?. This information is currently misplaced in the results sections, whereas it should be on the materials and methods section.

Response:

Thank you for your insightful comment. We have revised the manuscript to include detailed information regarding the human samples adnis also copied below for your consideration

Sample collection

A total of 30 gestational diabetes mellitus (GDM) patients, 30 IDA pregnant women and 17 healthy pregnant women were included in this research. Samples were collected from private clinics across various regions of Pakistan, including Lahore, Islamabad, Karachi, and Gujranwala, between January 2024 and March 2025. Participants were selected within the age range of 21 to 35 years to ensure homogeneity and relevance to the study objectives. All participants diagnosed with GDM were in the third trimester of pregnancy, specifically between 24 and 28 weeks of gestation. Written informed consent was acquired from all study participants. Patients with a history of pregestational Type 1 or Type 2 diabetes, chronic kidney disease, thyroid disorders, liver disease, or autoimmune diseases or other types of cancer were not considered for further investigation. Peripheral blood (5 mL) was drawn EDTA vacutainers while serum was isolated in a plastic vials and immediately stored at -80°C for preservation. This study was approved by the Institutional Ethics Committee, Institute of Molecular Biology and Biotechnology (IMBB), The University of Lahore (Project: PR/AI/GDM-1, Date 10-1-2025).

Comment:

Also, demographics are not described for the patient’s samples.

Response:

We are thankful to the reviewer for his valuable comments. We have now provided the demographic details of the patients sample in the revised manuscript

Comment:

The introduction needs to be improved. At the moment it doesn’t flow. Organize the ideas. Also include other diseases during pregnancy for which there is already evidence, for instance ectopic pregnancy and preeclampsia.

Response:

Thank you for your valuable feedback. We have revised the introduction to improve its logical flow and coherence. The ideas have been reorganized to provide a clearer progression from general background information to the specific focus of our study. Additionally, we have included references to other pregnancy-related conditions, such as ectopic pregnancy and preeclampsia, where relevant microRNA involvement has already been demonstrated. These additions help to better contextualize our research within the broader field of pregnancy-associated disorders.

Comment:

Data analysis needs to be improved as several graphs need to be analyzed by t-test, not by ANOVA

Response:

We appreciate the reviewer’s insightful comment. In response, we have consulted RNA expression analysis experts and re-evaluated our findings by conducting experiments in triplicates. Figure 2 has been accordingly revised.

Comment:

There are missing crucial references that need to be discussed such as:

https://onlinelibrary.wiley.com/doi/10.1155/2017/8067972

https://pubmed.ncbi.nlm.nih.gov/38404088/

https://www.frontiersin.org/journals/endocrinology/articles/10.3389/fendo.2022.892587/full

Response: We have now incorporated all of these references in the revised manuscript and are copied below for your consideration

  1. Masete, M., et al., A big role for microRNAs in gestational diabetes mellitus.Frontiers in Endocrinology, 2022. 13: p. 892587.
  2. Ali, Z., et al., Micro RNA 182-3-p, 519-d-5p, 378-3p as non-invasive predictors of preeclampsia.Journal of Ayub Medical College Abbottabad, 2023. 35(3).
  3. Barchitta, M., et al., The role of miRNAs as biomarkers for pregnancy outcomes: a comprehensive review.International Journal of Genomics, 2017. 2017(1): p. 8067972.

Comment:

Comments on the Quality of English Language

In general, the document is full of grammar mistakes, duplicated words, extra spaces. Formatting is required throughout the document, there are several Font types, sizes and even colors. All through the document there are several sentences with a mix or plural and singular, such as “selected miRNAs played significant role” (significant roles OR a significant role). Remove bold text throughout.

Response:

We have completed the proofreading of the manuscript. If further improvements are needed, we are open to consulting professional proofreading services.

Reviewer 3 Report

Comments and Suggestions for Authors

In this study, Badr Alzahrani et al. combined bioinformatics and RT-PCR validation to explore the patterns and regulatory targets of miR-182-3p and miR-24-3p in gestational diabetes mellitus (GDM) and iron deficiency anemia (IDA) during pregnancy. To make the review more integral, here are some issues that should be addressed.

  1. What is the main reason for selecting miR-182-3p and miR-24-3p over other miRNAs associated with GDM or IDA?
  2. GSE46287 comes from a rat model which may not properly reflect human conditions, is there any reason for choosing this dataset?
  3. How do the authors address potential batch effects or normalization issues when integrating GEO data with miRNA target prediction?
  4. Could the authors provide actual expression values (mean ± SD) for miR-182-3p and miR-24-3p in the different groups?
Comments on the Quality of English Language
  1. The paper is requiring English language copy editing. Please address this in your revised manuscript to ensure the best possible version of your manuscript. For example: Few studies reported that IDA is associted with reduced risk of GDM”. It can be revised as: " A few studies reported that IDA is associated with a reduced risk of GDM." Similarly, the sentence: " It is also accepted that GDM can lead to increase risk of neonatal death in mothers with GDM than nondiabetic mother." It can be revised as: " It is also accepted that GDM can lead to an increased risk of neonatal death in mothers with GDM compared to non-diabetic mothers."

Author Response

Dear Editor,

We wish to express our appreciation for your in-depth comments, suggestions, and corrections, and we would like to convey our sincere thanks for allowing us to improve our manuscript entitled “Combined Bioinformatic and Experimental Approaches to Analyze miR-182-3p and miR-24-3p Expression and Their Target Genes in Gestational Diabetes Mellitus and Iron Deficiency Anemia during pregnancy”.

We greatly appreciate the reviewers’ efforts in carefully reviewing our paper and for offering salient suggestions, which were of great help to us during this revision. We worked hard to be responsive to them.

We believe that we exhaustively resolved all points raised by the referees. We hope that the esteemed referees will find our responses satisfactory, and we are willing to include any future suggestions that the referees might have. Below is an abridged summary of each of the comments with a detailed response. We hope our revision meets your approval. Should you find the paper requires further clarification or revision, we most certainly stand ready to do so.

Looking forward to your positive response

Sincerely,

Aqsa Ikram (PhD)

Assistant Professor

Institute of Molecular Biology and Biotechnology,

University of Lahore.

Reviewer 1:

Comment:

What is the main reason for selecting miR-182-3p and miR-24-3p over other miRNAs associated with GDM or IDA?

Response:

Thank you for your thoughtful question. The selection of miR-182-3p and miR-24-3p was based on a combination of literature evidence and their reported involvement in key biological processes. We have included a brief justification in the revised manuscript under the Introduction section and is copied below for your consideration.

“MicroRNAs (miRNAs) are small non coding RNA and are crucial component of gene regulatory network and play a critical role in gene regulation [7].  Evidence from previous studies indicates that miR-182-3p is linked to tumor initiation and development, largely through its regulatory impact on cell proliferation and migration[8]. Suppression of miR-182-3p was found to mitigate the progression of GDM via regulation of INSR1, indicating its promise as a potential therapeutic target [9]. Significantly high expression levels of miR-182-3p is associated with preeclampsia (PE) [10]. Another investigation provides a fresh, integrative perspective on the role of miR-24-3p in the physiological regulation of RBP4, particularly in the context of trophoblast dysfunction and PE development[11]. It was also found to be highly expressed in women during the third trimester who later developed preeclampsia (PE). miRNA are also emerged as a major contributor to human diseases including GDM and IDA in pregnant women [7]. Multiple studies have demonstrated that changes in the expression of the maternal or placental miRNAs can indicate not only gestational complications—such as preeclampsia, spontaneous abortion, preterm birth, low birth weight, and macrosomia—but also prenatal exposure to environmental pollutants[12, 13].  Among these, miR-182-3p and miR-24-3p have emerged as potential regulators of genes involved in metabolic and hematological pathways. Inhibition of miR-182-3p has been found to be corelated with the greater likelihood of GDM and adverse pregnancy outcomes, but its role in IDA is not fully understood[14] [9].  While circulating miR-24-3p is observed to be linked  patients with previous GDM [15]. Similarly, its role in pregnant women with iron deficiency anemia (IDA) remains unclear.   Despite their predicted involvement, the precise roles of these miRNAs in pregnancy-related complications including GDM and IDA remain poorly defined”

Comment:

GSE46287 comes from a rat model which may not properly reflect human conditions, is there any reason for choosing this dataset?

Response:

We are thankful to the reviewer for his insightful comment. In the GSE dataset, we did not find any studies specifically related to IDA in pregnant women. Therefore, we initially used a rat model of IDA. However, to enhance the relevance of our study to human targets, we have revised the manuscript to include all genes associated with IDA through data mining. Specifically, we utilized multiple tools, including GeneCards, MalaCards, and the OMIM dataset, which collectively report IDA-specific genes based on more than 12,000 published studies.

Comment:

How do the authors address potential batch effects or normalization issues when integrating GEO data with miRNA target prediction?

Response:

We are thankful to the reviwer for his insghtful comments.  For GEO data  DEG with LogFC greater than 1 and a P-value less than 0.05 were included. In miRNA target genes we have taken only those genes with statistically significant values (>0.05).

Comment:

Could the authors provide actual expression values (mean ± SD) for miR-182-3p and miR-24-3p in the different groups?

Response:

In the present study

We calculate the  ΔCt

ΔCt=Cttarget​−Ctreference​

Then Calculated ΔΔCt

ΔΔCt=ΔC−ΔCtcontrol

Then fold change was claculated

Fold Change=2−ΔΔCt

miRNA

Mean±SD

Median

miR-182-3p

GDB

HC

2.052±0.7791

1.13±0.09

2.15

1.14

miR-24-3p

GDB

HC

3.2±0.59

0.99±0.58

3.9

0.4

miR-182-3p

IDA

HC

2.27±0.6

1.2 ±0.36

2.33

1.23

miR-24-3p

IDA

HC

1.9±0.86

0.9±0.38

1.78

0.82

Comment:

The paper is requiring English language copy editing. Please address this in your revised manuscript to ensure the best possible version of your manuscript. For example: Few studies reported that IDA is associted with reduced risk of GDM”. It can be revised as: " A few studies reported that IDA is associated with a reduced risk of GDM." Similarly, the sentence: " It is also accepted that GDM can lead to increase risk of neonatal death in mothers with GDM than nondiabetic mother." It can be revised as: " It is also accepted that GDM can lead to an increased risk of neonatal death in mothers with GDM compared to non-diabetic mothers."

Response:

We thank the reviewer for their valuable comments and have incorporated the suggested corrections. We have completed the proofreading of the manuscript. If further improvements are needed, we are open to consulting professional proofreading services. 

Reviewer 4 Report

Comments and Suggestions for Authors

Gestational diabetes mellitus (GDM) is characterized by elevated blood sugar levels first detected during pregnancy and is relatively prevalent worldwide. It is primarily caused by placental hormones that increase insulin resistance, rather than a deficiency in insulin production. Iron deficiency anemia (IDA), on the other hand, occurs when iron levels are insufficient to support healthy hemoglobin production—often due to the increased iron demands of the developing fetus and the expansion of maternal blood volume. Both GDM and IDA pose serious health risks for both mother and baby: GDM is associated with complications such as fetal overgrowth and neonatal hypoglycemia, while IDA can contribute to preterm birth, low birth weight, and delayed neurodevelopment. These risks emphasize the importance of early diagnosis, adequate nutritional support, and individualized care during pregnancy.

This manuscript investigates the roles of miR‑182‑3p and miR‑24‑3p in GDM and IDA. By combining RT‑PCR validation in clinical samples with in silico target prediction and differential gene expression analysis from GEO datasets, the authors identify overlapping target genes enriched in insulin signaling, apoptosis, and inflammatory pathways. Protein–protein interaction (PPI) network analysis highlights several hub genes as central regulators. The manuscript will make a valuable contribution to understanding miRNA‑mediated mechanisms in GDM and IDA, is well-referenced, and holds great relevance for patient care.

Comments:

Figure 1 Role(s) of miR-182-3p and miR-24-3p. A brief legend or call‑out box would improve interpretability. Key methods (e.g., which GEO datasets fed into which analyses) aren’t called out here.

Figure 2 The caption doesn’t specify whether bars represent SEM or SD. This is critical for assessing variability. While asterisks denote p‑values, the exact test used (e.g., ANOVA vs t‑test) and correction method aren’t mentioned in the figure legend. P‑value thresholds should also be defined beneath the plot.

Figure 3 The text refers to GSE154414 in Methods but Figure 3 captions mention GSE154413. Although DEGs are colored, the horizontal (p‑value) and vertical (log₂FC) cutoff lines are not drawn. Their absence makes it hard to judge which genes just meet vs far exceed thresholds

Figure 4 “Molecular process-related GO terms(C)cellular related GO terms (D)KEGG pathway”? Many of the top 10 terms are very broad. Highlighting a few more specific pathways or including a “zoom‐in” inset for IDA vs GDM differences could enhance biological insight.

Figure 5 In the full‐network view, most node labels overlap or are too small to read. Present either a zoomed‐in panel for hub nodes or list hubs in a table beside the figure. No quantitative network parameters (e.g., degree centrality, clustering coefficient) are provided. A small inset table summarizing these metrics for the highlighted hub genes would strengthen the analysis.

“cestine type endopeptidase activity” (p. 10) should read “cysteine‑type endopeptidase activity?

Keywords: .....differential gene expression (DEG)

“Furthermore, hub genes such as IRS1, PIK3CA, 24 CASP3, MAPK(7?), and PDGFRB”

..deficiency anemia through geo (GEO) analysis..

“with type 2 diabetes in patients with previous GDM [10].” Why is this sentence in bold?

Author Response

Dear Editor,

We wish to express our appreciation for your in-depth comments, suggestions, and corrections, and we would like to convey our sincere thanks for allowing us to improve our manuscript entitled “Combined Bioinformatic and Experimental Approaches to Analyze miR-182-3p and miR-24-3p Expression and Their Target Genes in Gestational Diabetes Mellitus and Iron Deficiency Anemia during pregnancy”.

We greatly appreciate the reviewers’ efforts in carefully reviewing our paper and for offering salient suggestions, which were of great help to us during this revision. We worked hard to be responsive to them.

We believe that we exhaustively resolved all points raised by the referees. We hope that the esteemed referees will find our responses satisfactory, and we are willing to include any future suggestions that the referees might have. Below is an abridged summary of each of the comments with a detailed response. We hope our revision meets your approval. Should you find the paper requires further clarification or revision, we most certainly stand ready to do so.

Looking forward to your positive response

Sincerely,

Aqsa Ikram (PhD)

Assistant Professor

Institute of Molecular Biology and Biotechnology,

University of Lahore.

Comment:

Figure 1 Role(s) of miR-182-3p and miR-24-3p. A brief legend or callout box would improve interpretability. Key methods (e.g., which GEO datasets fed into which analyses) aren’t called out here.

Response: Thank you for your valuable suggestion. We have revised Figure 1 to include a brief legend/call-out box summarizing the proposed roles of miR-182-3p and miR-24-3p, which we believe enhances the clarity and interpretability of the figure. These details have been incorporated into the figure legend and relevant sections of the main text for better transparency.

Comment:

Figure 2 The caption doesn’t specify whether bars represent SEM or SD. This is critical for assessing variability. While asterisks denote pvalues, the exact test used (e.g., ANOVA vs ttest) and correction method aren’t mentioned in the figure legend. Pvalue thresholds should also be defined beneath the plot.

Response:

We thank the reviewer for pointing out this important detail. The error bars in Figure 2 represent the SD, and this has now been explicitly stated in the figure legend. Additionally, we have clarified the statistical test used and included the method of multiple comparison correction, where applicable. P-value thresholds corresponding to asterisks (e.g., p<0.05, *p<0.01, etc.) have also been defined beneath the plot in the updated figure legend. In response, we have consulted RNA expression analysis experts and re-evaluated our findings by conducting experiments in triplicates. Figure 2 has been accordingly revised.

Comment:

Figure 3 The text refers to GSE154414 in Methods but Figure 3 captions mention GSE154413. Although DEGs are colored, the horizontal (pvalue) and vertical (log₂FC) cutoff lines are not drawn. Their absence makes it hard to judge which genes just meet vs far exceed thresholds

Response:

We are thankful to the reviwer comment. We havenow corrected our mistake in the revisedmanuscript.the horizontal (pvalue) and vertical (log₂FC) cutoff lines are now drawnin the revised manuscript.

Comment:

Figure 4 “Molecular process-related GO terms(C)cellular related GO terms (D)KEGG pathway”? Many of the top 10 terms are very broad. Highlighting a few more specific pathways or including a “zoom‐in” inset for IDA vs GDM differences could enhance biological insight.

Response: Thank you for your valuable suggestion.The GO analysis was done based on the interacting genes common among all the datasets. All DEG common among all the data sets and miRNA interacting genes were selected for further analysis. 351 genes were found to be common in all selected databases and miRNAs.

Comment:

Figure 5 In the full‐network view, most node labels overlap or are too small to read. Present either a zoomed‐in panel for hub nodes or list hubs in a table beside the figure. No quantitative network parameters (e.g., degree centrality, clustering coefficient) are provided. A small inset table summarizing these metrics for the highlighted hub genes would strengthen the analysis.

Response:

Thank you for your insightful feedback. We agree that the full-network view in Figure 5 is dense, making it difficult to read node labels clearly. To address this, we have added a zoomed-in panel focusing on the key hub nodes to enhance visibility and interpretability.

Comment:

“cestine type endopeptidase activity” (p. 10) should read “cysteinetype endopeptidase activity?

Response: We are thankful to the reviewer for his comment. We have no corrected our mistake in the revised manuscript.

Comment:

Keywords: .....differential gene expression (DEG)

“Furthermore, hub genes such as IRS1, PIK3CA, 24 CASP3, MAPK(7?), and PDGFRB”

..deficiency anemia through geo (GEO) analysis..

Response:

We are thankful to the reviewer for his helpful comments. We have now corrected our typos error in the revised manuscript.

Comment:

“with type 2 diabetes in patients with previous GDM [10].” Why is this sentence in bold?

Response: We are thankful to the reviewer for his comment. We have no corrected our mistake in the revised manuscript.

Round 2

Reviewer 3 Report

Comments and Suggestions for Authors

The authors have addressed all of my previously raised concerns. However, I still recommend that the manuscript undergo further English language improvement and professional proofreading to enhance clarity, grammar, and overall readability.

Comments on the Quality of English Language

I still recommend that the manuscript undergo further English language improvement and professional proofreading to enhance clarity, grammar, and overall readability.

Author Response

Response letter:

Dear Editor,

We wish to express our appreciation for your in-depth comments, suggestions, and corrections, and we would like to convey our sincere thanks for allowing us to improve our manuscript entitled “Combined Bioinformatic and Experimental Approaches to Analyze miR-182-3p and miR-24-3p Expression and Their Target Genes in Gestational Diabetes Mellitus and Iron Deficiency Anemia during pregnancy”.

We greatly appreciate the reviewers’ efforts in carefully reviewing our paper and for offering salient suggestions, which were of great help to us during this revision. We worked hard to be responsive to them.

We are grateful for your positive remarks regarding the English language improvement, which have encouraged us greatly. Your feedback has strengthened the manuscript, and we are pleased that our work aligns with your expectations.

Reviewer 3:

Comment:

The authors have addressed all of my previously raised concerns. However, I still recommend that the manuscript undergo further English language improvement and professional proofreading to enhance clarity, grammar, and overall readability.

Response:

Regarding the recommendation for further English language improvement and professional proofreading, we fully agree with your suggestion. Accordingly, we have carefully revised the manuscript to enhance its clarity, grammar, and overall readability. Additionally, we have utilized a professional language editing service to ensure the manuscript meets high linguistic standards (document attached).
